# Striking Neurochemical and Behavioral Differences in the Mode of Action of Selegiline and Rasagiline

**DOI:** 10.3390/ijms241713334

**Published:** 2023-08-28

**Authors:** Laszlo G. Harsing, Julia Timar, Ildiko Miklya

**Affiliations:** Department of Pharmacology and Pharmacotherapy, Semmelweis University, Nagyvarad ter 4, 1089 Budapest, Hungary; harsing.laszlo@med.semmelweis-univ.hu (L.G.H.J.); timar.julia@med.semmelweis-univ.hu (J.T.)

**Keywords:** selegiline, rasagiline, dopaminergic activity enhancer effect, trace amine-associated receptor 1, monoamine oxidase inhibition, [^3^H]dopamine release, rat striatum, conditioned avoidance response

## Abstract

Selegiline and rasagiline are two selective monoamine oxidase B (MAO-B) inhibitors used in the treatment of Parkinson’s disease. In their clinical application, however, differences in L-dopa-sparing potencies have been observed. The aim of this study was to find neurochemical and behavioral explanations for the antiparkinsonian effects of these drugs. We found that selegiline possesses a dopaminergic enhancer effect: it stimulated the electrically induced [^3^H]dopamine release without influencing the resting [^3^H]dopamine release from rat striatal slices in 10^−10^–10^−9^ mol/L concentrations. Rasagiline added in 10^−13^ to 10^−5^ mol/L concentrations did not alter the resting or electrically stimulated [^3^H]dopamine release. Rasagiline (10^−9^ mol/L), however, suspended the stimulatory effect of selegiline on the electrically induced [^3^H]dopamine release. The trace amine-associated receptor 1 (TAAR1) antagonist EPPTB (10^−8^–10^−7^ mol/L) also inhibited the stimulatory effect of selegiline on [^3^H]dopamine release. The effect of selegiline in its enhancer dose (5.33 nmol/kg) against tetrabenazine-induced learning deficit measured in a shuttle box apparatus was abolished by a 5.84 nmol/kg dose of rasagiline. The selegiline metabolite (−)methamphetamine (10^−9^ mol/L) also exhibited enhancer activity on [^3^H]dopamine release. We have concluded that selegiline acts as an MAO-B inhibitor and a dopaminergic enhancer drug, and the latter relates to an agonist effect on TAAR1. In contrast, rasagiline is devoid of enhancer activity but may act as an antagonist on TAAR1.

## 1. Introduction

More than sixty years ago, Joseph Knoll started a research project on newly synthesized phenylethylamine derivatives. Among them was selegiline ((−)deprenyl, N-methyl-*N*-(2-propinyl)-2-methyl-1-phenylethylamine), and the compound later became the first known selective B-type monoamine oxidase (MAO) inhibitor [1]. The possible therapeutic use of selegiline was obscure until the mid-seventies, when it became evident that the drug exerts an L-dopa-sparing effect in Parkinson’s disease therapy [2,3]. From this point of view, it was an essential finding that dopamine, the major neurotransmitter that is lost in the parkinsonian extrapyramidal system [4], is the substrate for MAO-B in humans [5]. Although selegiline became a standard element of the treatment of Parkinson’s disease, pitfalls of the drug arose when its metabolism was investigated, leading to the demonstration that L-methamphetamine and L-amphetamine are among the metabolites [6]. Since the propargyl group in selegiline was considered responsible for the MAO inhibition, novel chemical synthetic works have been initiated to obtain compounds containing the propargyl side chain attached to other aryl groups. A more extended alteration in the chemical structure of selegiline resulted in the compound J-508 as a novel MAO-B inhibitor [7]. Later, desmethyl J-508 became the therapeutically important antiparkinsonian MAO-B inhibitor, rasagiline (Figure 1) [8,9].

It is generally accepted that the therapeutic potential of selegiline in the treatment of Parkinson’s disease is due to the inhibition of oxidative deamination of dopamine, the key neurotransmitter in the extrapyramidal system. In addition to this, a series of additional pharmacologic actions with potential importance in the therapeutic profile of selegiline have been demonstrated. Thus, selegiline inhibits dopamine reuptake into dopaminergic axon terminals and increases dopamine release from rat striatal slices [11,12]. These two effects were believed to play a role in the inhibition of acetylcholine release from striatal cholinergic interneurons [11]. 

In search of selegiline’s mode of action, a series of analog compounds have been synthesized. Knoll and co-workers [13,14] reported the pharmacological actions of some aryl-substituted 2-propyl-aminopentanes on catechol- and indoleamine neurochemical transmission in the central nervous system. These compounds were capable of increasing the electrically stimulated dopamine, noradrenaline, and serotonin release in low femto/picomolar concentrations in in vitro conditions without affecting the resting release of these biogenic amines [15]. Compounds exhibiting these unique pharmacological actions are designated as catecholaminergic activity enhancer (CAE) drugs, and the most widely investigated compounds of this series are (−)PPAP ((−)-1-phenyl-2-propylaminopentane, Figure 1), and its indole (−)IPAP ((−)-1-(indol-3-yl)-2-propylaminopentane), and benzofuran ((−)BPAP ((*R*)-(−)-1-(benzofuran-2-yl)-2-propylaminopentane) analogs [14]. An important breakthrough was the demonstration that selegiline but not rasagiline exhibits enhancer effects in a number of behavioral tests [16,17].

Recently, we have concluded that the enhancer compound (−)BPAP increased the electrically induced [^3^H]dopamine release from the striatum via activation of the trace amine-associated receptor 1 (TAAR1) [18]. TAAR1 has a central role in the regulation of membrane potential, transporter operation, non-vesicular and vesicular neurotransmitter release, and presynaptic autoreceptor-mediated feedback inhibition [19,20]. Integration of these neurochemical events at axon terminal levels has led to the conclusion that the CAE effect of the enhancer compounds is mediated, at least in part, by activation of the TAAR1 signal transduction pathway.

Several MAO-B inhibitors have been tested in different behavioral methods examining the cognitive function of animals, for example, pot jumping performance for motor skill learning, a Morris water-maze for spatial learning, a cooperation task for social cognition, object recognition, passive and active avoidance (shuttle box) tests for the examination of attention, fear memory, and learning ability [21]. The latter procedure was employed in our current study.

The aim of the present study was to obtain further details about the pharmacological actions of selegiline as an enhancer and MAO-B inhibitor, and rasagiline, which is devoid of a CAE effect but selectively inhibits the MAO-B enzyme. For this, we extended our investigation into the interaction of dopaminergic enhancer activity and TAAR1 signaling. Further on, we compared some behavioral effects of selegiline and rasagiline and the consequences of their combination.

## 2. Results

### 2.1. Neurochemical Studies

#### 2.1.1. [^3^H]Dopamine Content and Release in Superfused Striatum of the Rat

The [^3^H]dopamine tissue content in the striatal slices was 390.59 ± 111.71 kBq/g after the 60 min initial washout period. This content of radioactivity decreased to 232.60 ± 58.03 kBq/g in the following a 75-min-long superfusion. Thus, approximately 40 percent of the striatal tissue’s [^3^H]dopamine content was released during the superfusion (Figure 2).

The resting [^3^H]dopamine release from the superfused striatal slices reached a value of 6.00 ± 1.78 kBq/g/3 min after the 60-min-long preperfusion. Electrical stimulation increased this release to 12.72 ± 4.00 kBq/g/3 min. These values of [^3^H]dopamine release corresponded to 1.59 ± 0.07 and 3.59 ± 0.64 percent of the tissue’s [^3^H]dopamine content released in 3 min (fractional release, n = 4, *p* < 0.01). 

#### 2.1.2. Effect of β-Phenylethylamine and the MAO Inhibitors Selegiline and Rasagiline on Resting [^3^H]Dopamine Release in the Striatum

The trace amine β-phenylethylamine, added in a concentration of 10^−5^ mol/L, evoked [^3^H]dopamine release from superfused rat striatal slices (Figure 3A). In contrast to electrical stimulation, this release is considered to be an external Ca^2+^-independent process, originating from cytoplasmic neurotransmitter pools, and is a result of the reverse-mode operation of the plasma membrane dopamine transporter [22,23]. Non-vesicular neurotransmitter release is typically induced by trace amines and releaser substances (i.e., amphetamines and related compounds).

Figure 3B shows that selegiline and rasagiline, in which the bulky propargyl group substitution is added to the nitrogen atom, failed to evoke non-vesicular [^3^H]dopamine release, whereas this chemical alteration led to MAO-B inhibitory properties. This effect of the selective MAO-B inhibitors is in contrast to that of the trace amines or the releaser amphetamines, which induce an increase in resting [^3^H]dopamine release [18].

#### 2.1.3. The Dopaminergic Activity Enhancer Effect of Selegiline

To investigate whether selegiline possesses a dopaminergic activity enhancer effect, the drug was added to striatal slices in increasing concentrations. As shown in Figure 4A, selegiline in a concentration range of 10^−13^ to 10^−5^ mol/L elicited a biphasic effect: it increased the electrically stimulated [^3^H]dopamine release in concentrations of 10^−10^ and 10^−9^ mol/L and in concentrations of 10^−6^ and 10^−5^ mol/L, respectively. The former effect was considered a specific enhancer effect.

While selegiline enhanced the electrical stimulation-induced [^3^H]dopamine release in low and high concentrations, it was without effect on the resting release in a concentration range of 10^−13^ to 10^−5^ mol/L (Figure 4B). Thus, selegiline may primarily influence dopamine release from vesicular stores, and it exerts only a limited effect on non-vesicular release; these are characteristics of the enhancer compounds [18].

#### 2.1.4. EPPTB Reversed the Stimulatory Effect of Selegiline on [^3^H]Dopamine Release in Rat Striatum

We tested the effect of EPPTB, the first described selective inhibitor of TAAR1 [24], on the dopaminergic activity enhancer effect of selegiline in a rat striatum. Previously, we reported that EPPTB *per se* did not influence resting or electrical stimulation-induced [^3^H]dopamine release the in rat striatum when added in a 10^−8^ or 10^−7^ mol/L concentration [18]. This suggests that TAAR1 does not possess constitutive activity in striatal dopaminergic neurochemical transmission. As shown in Figure 5, however, EPPTB suspended the dopaminergic enhancer effect of selegiline on the electrical stimulation-induced [^3^H]dopamine release from the rat striatum.

#### 2.1.5. Effect of Rasagiline on [^3^H]Dopamine Release in Rat Striatum

To determine whether rasagiline, similarly to selegiline, also possesses a dopaminergic activity enhancer effect, the drug was added to striatal slices in increasing concentrations. Figure 6A shows that rasagiline failed to enhance the electrical stimulation-induced release of [^3^H]dopamine in a concentration range of 10^−13^ to 10^−5^ mol/L The lack of effect of rasagiline on the electrical stimulation-induced [^3^H]dopamine release was in contrast to that of selegiline. Rasagiline was also without effect on the resting [^3^H]dopamine release in rat striatum when added in a concentration range of 10^−13^ to 10^−5^ mol/L (Figure 6B).

#### 2.1.6. Interaction of Rasagiline and Selegiline on [^3^H]Dopamine Release in Rat Striatum

[^3^H]Dopamine release was measured from rat striatum in the presence of selegiline, rasagiline, or their combination (Figure 7). As was expected, the selegiline increased, and rasagiline was without effect on the electrical stimulation-evoked release of [^3^H]dopamine in this series of experiments. When the effect of selegiline was determined on the electrical stimulation-induced [^3^H]dopamine release in the presence of rasagiline, no dopaminergic activity enhancer effect of selegiline was detected.

#### 2.1.7. Effect of the Metabolites of Selegiline and Rasagiline on [^3^H]Dopamine Release in Rat Striatum

Figure 8 demonstrates that the despropargyl metabolite of selegiline, (−)methamphetamine, exhibited dopaminergic enhancer activity in the rat striatum. (−)Methamphetamine added in a concentration of 10^−9^ mol/L increased the electrically induced [^3^H]dopamine release without altering the resting release. This effect of (−)methamphetamine resembles the selegiline effect (10^−9^ mol/L) on the resting and electrical stimulation-induced [^3^H]dopamine release (Figure 4).

In contrast to the selective B-type MAO inhibitors selegiline and rasagiline, (−)methamphetamine increased the resting [^3^H]dopamine release when added in a concentration of 10^−5^ mol/L (Figure 8B). This effect of (−)methamphetamine was markedly reduced compared to that of (±)methamphetamine. The despropargyl analog of rasagiline, (−)-1-aminoindane, was without effect on the resting striatal dopamine release.

### 2.2. Behavioral Studies

#### Interaction of Rasagiline and Selegiline on Shuttle Box Test

As Figure 9 shows, selegiline in its specific enhancer dose (0.001 mg/kg equivalent to 5.33 nmol/kg sc.) significantly reduced the complete abolishment of the conditioned avoidance response (CAR) and the escape response (escape failure—EF) induced by tetrabenazine in a dose of 1 mg/kg. Rasagiline in the same “enhancer equivalent” dose (0.001 mg/kg equivalent to 5.84 nmol/kg sc.) failed to influence the effect of the tetrabenazine. However, the coadministration of rasagiline with selegiline totally terminated the enhancer effect of the selegiline.

## 3. Discussion

The aim of the present study was to investigate whether the two MAO-B inhibitors selegiline and rasagiline possess dopaminergic enhancer activity in rat striatum. Knoll and coworkers [14] reported that some aryl-substituted 2-propyl-aminopentanes were capable of increasing the electrical stimulation-evoked release of biogenic amines (dopamine, noradrenaline, serotonin) in femto/picomolar concentrations in in vitro conditions without altering the resting release of these biogenic amines [15]. Compounds exhibiting these unique pharmacological actions are designated as catecholamine activity-enhancer drugs. Using [^3^H]dopamine release measurements, we found that selegiline but not rasagiline exhibits enhancer activity. This observation may serve as neurochemical evidence for previous findings demonstrating the enhancer effect of selegiline in behavioral tests [17].

The dopaminergic enhancer effect of selegiline appeared in the low 10^−10^ and 10^−9^ mol/L concentrations, in which it increased the electrically stimulated [^3^H]dopamine release without affecting the resting release. Selegiline, however, also increased dopamine release in 10^−6^ and 10^−5^ mol/L concentrations; this effect may be related to the inhibition of A-type MAO activity [25]. Dopamine, in contrast to the human extrapyramidal system, is a MAO-A substrate in the rat striatum [26,27,28]. Previously, we measured the effect of selegiline on [^3^H]dopamine deamination in rat striatal slices and found a 76% inhibition at 10^−5^ mol/L concentration [12].

Of the related structures, we recently studied the N,α,-dipropylarylethylamine derivative (−)BPAP as an enhancer compound in dopaminergic neurotransmission of the rat striatum [18]. Previous investigations of other members of the N,α,-dipropylarylethylamine series ((−)PPAP, (−)IPAP) indicated that the catecholamine activity enhancer effect is not related to MAO inhibition [13,14]. Therefore, it may be a unique property of selegiline to exhibit both dopaminergic enhancer activity and MAO inhibitory effects.

The mechanism of how enhancer compounds are able to increase dopaminergic neurochemical transmission was obscure until this effect was not directly linked to TAAR1 [29]. We previously reported that EPPTB, the selective antagonist of TAAR1 [24], reversed the increase of electrical stimulation-induced [^3^H]dopamine release by (−)BPAP in the rat striatum [18]. This finding prompted us to study whether EPPTB also suspends the dopaminergic enhancer activity of selegiline. It was found that EPPTB, used in specific concentrations (10^−8^ and 10^−7^ mol/L), antagonized the enhancer activity of selegiline in the regulation of dopaminergic neurochemical transmission in the striatum. Our finding confirms previous conclusions that the therapeutic value of selegiline in the treatment of Parkinson’s disease may be dual, and the dopaminergic enhancer activity of this drug is complementary to its B-type MAO inhibitory effect [30].

Besides selegiline, rasagiline has also been extensively studied as a MAO-B inhibitor [31,32]. We targeted rasagiline to investigate whether it also exerts dopaminergic enhancer activity in our experimental conditions. In contrast to selegiline, however, low concentrations of rasagiline failed to stimulate the electrical stimulation-evoked [^3^H]dopamine release from rat striatal slices, indicating a lack of enhancer activity of this MAO-B inhibitor. The missing enhancer activity of rasagiline was also shown by behavioral results. While a specific enhancer dose of selegiline reduced the tetrabenazine-induced abolishment of learning ability, rasagiline failed to do so (Figure 9). These findings further support our view that the enhancer activity and MAO inhibition are two independent properties: the enhancer activity is present in the pharmacological profiles of (−)BPAP and selegiline, whereas both selegiline and rasagiline are capable of inhibiting MAO-B enzyme activity.

Further, we investigated the interaction of selegiline and rasagiline in the release of [^3^H]dopamine in the rat striatum. It turned out that rasagiline, which *per se* did not exhibit enhancer activity on dopaminergic neurochemical transmission, antagonized the enhancer effect of selegiline when added in equimolar concentrations. The same results were achieved in the behavioral study, too: the rasagiline terminated the enhancer effect of the selegiline against tetrabenazine (Figure 9). Assuming that the dopaminergic enhancer activity of selegiline was due to an agonist effect on TAAR1, we speculated that rasagiline acts as a TAAR1 signaling inhibitor, resulting in an antagonistic drug interaction (Figure 10). This concept needs, however, further clarification by using a TAAR1 binding assay or cAMP accumulation measurements in response to selegiline and rasagiline. We also speculated that the stereochemical differences between the substituted phenyl ring in selegiline and the substituted indane ring of rasagiline may be responsible for the different interactions of the two drugs with TAAR1. The limitation of our study is, however, the lack of direct evidence for the role of TAAR1 in the neurobiology of Parkinson’s disease.

The clinical use of the MAO inhibitors selegiline and rasagiline in the treatment of Parkinson’s disease is now solid. However, the antiparkinsonian effect of the catecholamine activity enhancer drugs is far less documented. It is worthwhile to mention the early clinical findings showing that (−)BPAP, which solely possesses enhancer activity without MAO inhibition, may have antiparkinsonian potential [39].

We also confirmed previous findings that (−)methamphetamine, one of the major metabolites of selegiline, possesses a dopaminergic enhancer effect in the striatum [5,40]. It is worthwhile to mention, however, that the enhancer effect of (−)methamphetamine appears in the low nanomolar concentrations, in which the dopamine releaser effect of the drug is still not shown [18].

Although rasagiline has gradually replaced selegiline in the treatment of Parkinson’s disease, clinical comparisons have indicated significant differences in the mode of action of these two drugs. Thus, an early clinical study of rasagiline in Parkinson’s patients failed to demonstrate a decreased need for levodopa, an effect opposite to that of selegiline [41]. A systematic meta-analysis of B-type MAO inhibitors for Parkinson’s disease revealed that selegiline, rasagiline, and safinamide also were effective in the treatment of the symptoms of Parkinson’s disease when given as a monotherapy. Among them, however, selegiline proved to be the most effective in combination with levodopa [42]. Furthermore, Peretz and coworkers [43] reported that the time from a MAO-B inhibitor prescription until the initiation of dopamine agonists or levodopa therapy was longer in Parkinson’s patients treated with selegiline in comparison with rasagiline treatment. Differences in the levodopa-sparing effects of rasagiline and safinamide have also been reported recently [44]. As rasagiline does not exhibit a CAE effect, in contrast to selegiline, this might explain why “…based on current evidence, rasagiline cannot be said to definitely have a disease-modifying effect” [45]. Whether or not these observed differences in the clinical actions of selegiline and rasagiline are due to the dopaminergic activity enhancer effect needs further elucidation.

## 4. Materials and Methods

### 4.1. Animals

We used male Wistar rats (Toxicoop, Hungary) weighing 180–220 g for our experiments. The animals were housed five per cage in a standard temperature- and humidity-controlled animal facility using a 12 h light/dark cycle (6.00 a.m. on, 6.00 p.m. off). Food and water were available *ad libitum*. All the experimental procedures were approved by local ethical committees and were carried out in accordance with the NIH Guide for the Care and Use of Laboratory Animals, 8th Edition, 2011.

### 4.2. Neurochemical Experiments

#### 4.2.1. Brain Slice Preparation

The rats were decapitated by a guillotine, and the brains were removed from the skulls. The striatum was dissected as described by Glowinski and Iversen [46], and striatal slices were prepared by using a McIlwain tissue chopper (Ted Pella Inc., Redding, CA, USA). After preparation at room temperature, the striatal slices were immersed in aerated (95% O_2_/5% CO_2_) Krebs bicarbonate buffer with a composition of NaCl 118, KCl 4.7, CaCl_2_ 1.25, NaH_2_PO_4_ 1.2, MgCl_2_ 1.2, NaHCO_3_ 25, glucose 11.5, ascorbic acid 0.3, and Na_2_EDTA 0.03 mmol/L.

#### 4.2.2. Release of [^3^H]Dopamine from Rat Striatal Slices

The striatal slices of the rat were incubated in the presence of 10 µCi of [^3^H]dopamine added to 1.5 mL oxygenated (95% O_2_/5% CO_2_, pH 7.4) and preheated (37 °C) Krebs bicarbonate buffer for 30 min [47]. The superfused striatal tissues were transferred into tissue chambers (in a volume of 0.3 mL), after loading with [^3^H]dopamine, and superfused with aerated and preheated Krebs bicarbonate buffer (Experimetria Kft, Budapest, Hungary). The flow rate was maintained at 1 mL/min by using a Gilson peristaltic pump (type M312, Villiers-Le Bel, France). The superfusate was discarded during the first 60 min preperfusion period, then, twenty-five fractions (3 min each) were collected by a fraction collector (Gilson type FC-203B, Middletown, WI, USA). Electrical field stimuli (parameters: 40 V voltage, 10 Hz frequency, 2 msec impulse duration for 3 min in fractions 4 and 18) were delivered by a Grass S88 Electrostimulator (Quincy, MA, USA) to induce vesicular [^3^H]dopamine release. The drugs were added to the striatal slices between the first and second electrical stimulations, as was indicated in the figure legends.

#### 4.2.3. Determination of [^3^H]Dopamine Efflux

The striatal tissues were collected from the superfusion chambers at the end of the superfusion, weighed, and solubilized in 0.4 mL of Soluene-350. An aliquot (50 µL) was vortex-mixed with a liquid scintillation reagent (5 mL) and subjected to liquid scintillation spectrometry in order to determine the content of radioactivity remaining in the tissues. The tissue content of [^3^H]dopamine was calculated as kBq/g tissue.

For determination of the [^3^H] radioactivity released from the brain slices, a sample of the collected superfusate (1 mL) was mixed with Soluene-350 liquid scintillation reagent (5 mL) and subjected to liquid scintillation spectrometry. The release of [^3^H]dopamine was expressed as kBq/g/3 min fraction or as a fractional rate, i.e., a percentage of the amount of radioactivity in the tissue at the time of the release was determined. To calculate the stimulation-evoked [^3^H]dopamine release, the mean of the basal outflow determined before and after stimulation was subtracted from each sample and summed [47].

The effects of the drugs on the [^3^H]dopamine efflux at rest were expressed by the B2/B1 ratio, i.e., [^3^H]dopamine outflow in fraction 17 (presence of drug, B2) and fraction 3 (absence of drug, B1). The effects of the drugs on the electrical stimulation-evoked [^3^H]dopamine release were determined with the ratio of [^3^H]dopamine release measured in response to the 2nd (presence of drug, S2) and 1st (absence of drug, S1) stimulations, i.e., the S2/S1 ratio (Figure 2A). The Quattro Pro and the GraphPad Prism computer programs were used for the data calculation.

### 4.3. Behavioral Experiment

To investigate the difference between rasagiline and the synthetic enhancer substance selegiline on the rat striatal neurons *in vivo*, we used a modified version of the shuttle box [48]. The tetrabenazine-induced learning deficit can be antagonized by a synthetic enhancer substance [17]. To test a compound’s ability to enhance the acquisition of conditioned avoidance reflex (CAR), it is necessary to select proper training conditions. In the shuttle box, the acquisition of a CAR was analyzed over 5 consecutive days with 100 trials daily. The rat was put in a box, which was separated by a barrier with a small gate inside. The rats were trained to cross the barrier when a conditioned stimulus (CS, light flash) was applied. If the animal did not enter the other side of the box within 15 s, it was punished with the unconditioned stimulus (US), a foot shock (1 mA). Escape failure (EF) was considered if the rat failed to respond to the US. A trial contained a 10 s intertrial interval (IR) and 20 s of CS. The last 5 s of the CS overlapped the 5 s US (Figure 11). The number of CARs, EFs, and IRs was automatically counted and evaluated. The shuttle box behavior was measured at the same time each day. Saline, the studied drugs, and tetrabenazine were given 30 min before starting the daily session.

The rats were divided into 5 groups and treated with saline (SAL), saline + tetrabenazine (SAL + TBZ), saline + tetrabenazine + selegiline (SAL + TBZ + SEL), saline + tetrabenazine + rasagiline (SAL + TBZ + RAS), and saline + tetrabenazine + selegiline + rasagiline (SAL + TBZ + SEL + RAS). The tetrabenazine was applied in a dose of 1 mg/kg sc, and the selegiline and rasagiline were injected sc in a dose of 0.001 mg/kg, respectively.

### 4.4. Statistical Analyses

The Student’s *t*-statistics for two-means, Student’s paired *t*-test, and one-way ANOVA followed by Dunnett’s test were used for the statistical analysis of the data. The two-way ANOVA and Bonferroni post hoc test were applied for analyzing the shuttle box results. The number of independent determinations was indicated by n, and the data were expressed as the mean ± S.E.M. The level of probability (*p*) was set at 5% for consideration significance.

### 4.5. Materials

Dihydroxyphenylethylamine-3,4[^3^H]([^3^H]dopamine) with specific activity of 27.8 Ci/mmol, Ultima Gold XR liquid scintillation reagent, and Soluene-350 tissue solubilizer were purchased from PerkinElmer Life and Analytical Sciences, Boston, MA, USA. Selegiline ((−)deprenyl, N-methyl-*N*-(2-propinyl)-2-methyl-1-phenylethylamine), and rasagiline (N-propargyl-(1-indenyl)-ammonium) were gifts from the Fujimoto Pharmaceutical Co., Osaka, Japan. EPPTB (Ro5212773, N-(3-ethoxyphenyl)-4-pyrrolidin-1-yl-3-trifluoro-methylbenzamide), β-phenylethylamine HCl (PEA), and (R)-(−)-1-aminoindane were purchased from Sigma-Aldrich Chemical Co, Budapest, Hungary. (−)Methamphetamine HCl and (±)methamphetamine HCl were donated by Chinoin Pharmaceuticals, Budapest, Hungary. The tetrabenazine HCl was synthesized by Prof. Dr. Csaba Szantay, Department of Organic Chemistry, Budapest University of Technology and Economics, Budapest, Hungary. All the other chemicals were of analytical grade.

## 5. Conclusions

In conclusion, our experiments provide evidence that selegiline possesses a dopaminergic activity enhancer effect in addition to MAO-B enzyme inhibition, and the two effects are independent. In contrast to selegiline, rasagiline solely inhibits MAO-B activity without possessing dopaminergic enhancer activity. We have speculated that the presence of the dopaminergic enhancer activity in the pharmacological spectrum of selegiline may be the basis of the clinical observations reported in the antiparkinsonian effects of selegiline in comparison with rasagiline. Whether the dopaminergic activity enhancer effect of selegiline is responsible for the differences in the levodopa-sparing effect of selegiline and rasagiline is currently being investigated in our laboratory.

## Figures and Tables

**Figure 1 ijms-24-13334-f001:**
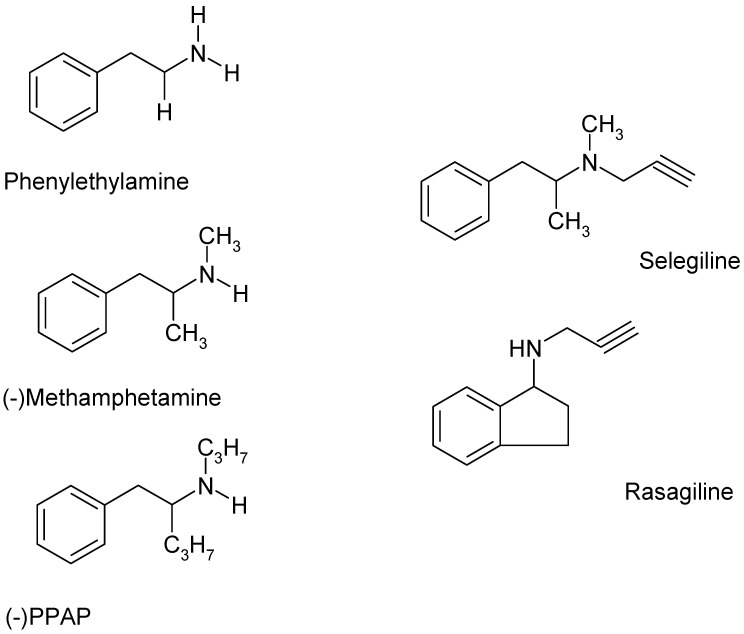
The chemical structures of the trace amine β-phenylethylamine, the releaser compound methamphetamine, the selective catecholamine activity enhancer (−)PPAP ((−)-1-phenyl-2-propylaminopentane), and the MAO-B inhibitors selegiline ((−)deprenyl) and rasagiline. Although the compounds shown here can be derived from β-phenylethylamine, there are definitive differences in the attached alkylamine side chains. In addition, a phenyl ring is present in the structures of β-phenylethylamine, methamphetamine, (−)PPAP, and selegiline, and there is an indane ring in rasagiline. (−)PPAP exerts a catecholamine activity enhancer effect, primarily on catecholaminergic transmission, but leaves MAO activity unchanged [10]. Selegiline and rasagiline, which possess a propargyl group attached to the nitrogen atom in the side chain, proved to be potent MAO-B inhibitors.

**Figure 2 ijms-24-13334-f002:**
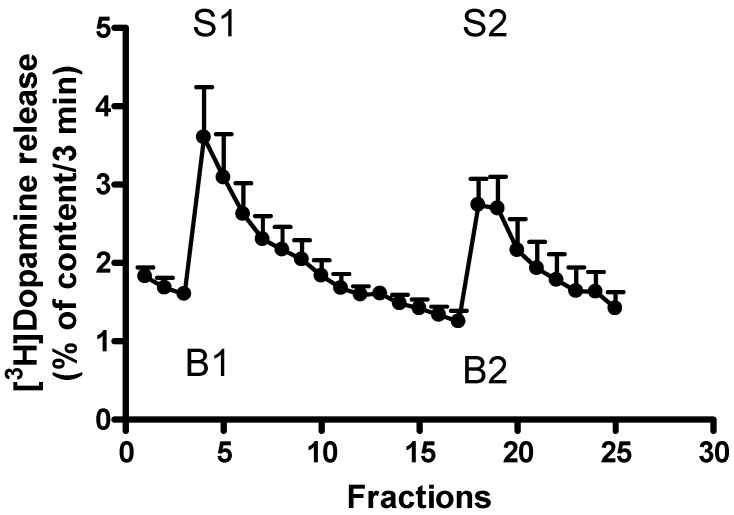
The time course of resting and electrical stimulation-induced [^3^H]dopamine release from rat striatum. Slices from rat striatum were prepared, incubated in the presence of [^3^H]dopamine, and superfused with oxygenated and preheated Krebs bicarbonate buffer. The [^3^H]dopamine release was expressed as the percent of content released, i.e., a percentage of the amount of [^3^H]dopamine in the tissue at the time of the release. The [^3^H]dopamine release was stimulated electrically (40 V, 10 Hz, 2 msec for 3 min) in fractions 4 (S1) and 18 (S2). The electrically stimulated fractional release S2 (2nd stimulation) over fractional release S1 (1st stimulation) (S2/S1 ratio) was 0.85 ± 0.06, representing a [^3^H]dopamine release with a vesicular origin. The resting fractional release B2 (fraction 17) over fractional release B1 (fraction 3) (B2/B1 ratio) was 0.77 ± 0.06. The tissue [^3^H]dopamine content approached a value of 232.60 ± 58.03 kBq/g at the end of the superfusion, mean ± S.E.M., n = 4. When the drug effect was studied, the drugs were added to the superfusion buffer between the 1st and 2nd electrical stimulations and maintained throughout the experiment.

**Figure 3 ijms-24-13334-f003:**
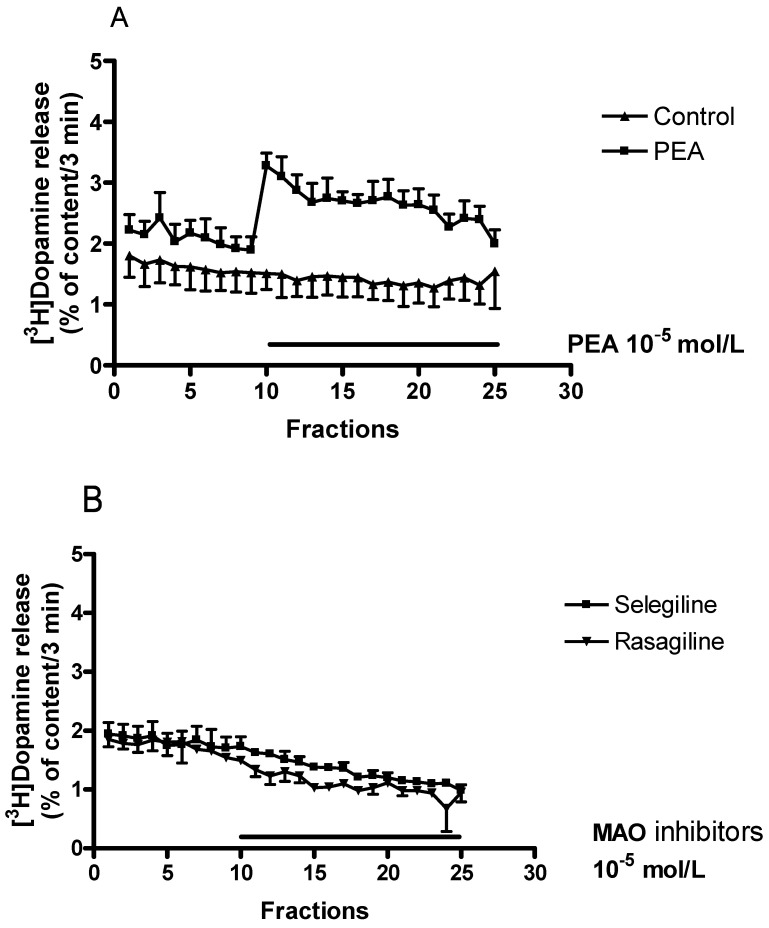
Figure 3 shows the time course of resting [^3^H]dopamine release in the presence and absence of drugs (β-phenylethylamine, selegiline, rasagiline) measured from the rat striatum. Striatal slices from a rat brain were prepared, loaded with [^3^H]dopamine, and superfused with aerated and preheated Krebs bicarbonate buffer. The release of [^3^H]dopamine was expressed as a fractional rate calculated as the percent of content released. (**A**) Non-vesicular [^3^H]dopamine release was evoked by β-phenylethylamine (PEA) added to the superfusion buffer from fraction 10 and maintained throughout the experiment. A quantity of 10^−5^ mol/L of β-phenylethylamine increased non-vesicular [^3^H]dopamine release from 1.89 ± 0.21 to 3.27 ± 0.20 percent of the content released in 3 min (fractions 9 and 10), mean ± S.E.M., n = 4–5. (**B**) The effects of the MAO B inhibitors selegiline and rasagiline on resting [^3^H]dopamine release from rat striatum. For control, see Figure 3A. Selegiline and rasagiline were added to striatal slices from fraction 10 in a concentration of 10^−5^ mol/L and kept present throughout the experiment. In contrast to β-phenylethylamine, neither selegiline nor rasagiline altered the resting [^3^H]dopamine release in striatal slices, mean ± S.E.M., n = 4.

**Figure 4 ijms-24-13334-f004:**
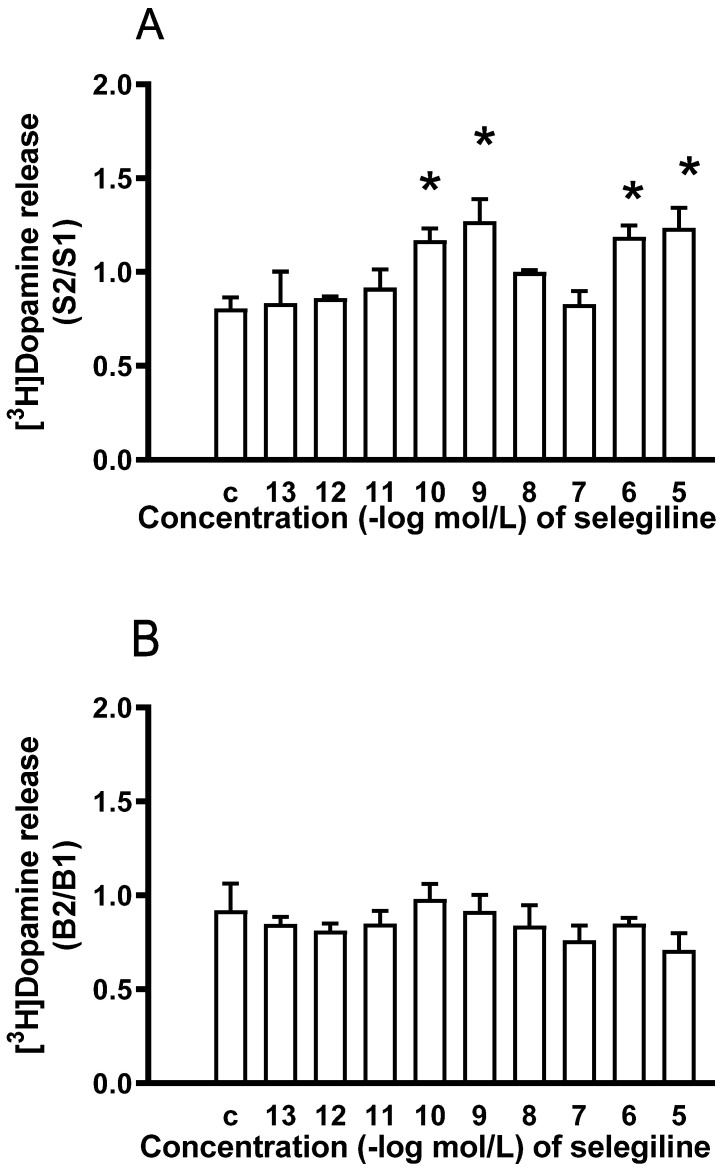
Concentration-dependent effects of selegiline on resting and electrical stimulation-induced [^3^H]dopamine release from rat striatum. For experimental procedure, see Figure 2. The resting and the electrical stimulation-induced [^3^H]dopamine release were determined as fractional rates. Selegiline was added in a concentration range from 10^−13^ to 10^−5^ mol/L to striatal slices. (**A**) The effect of selegiline on the electrical stimulation-induced [^3^H]dopamine release determined in the 1st (absence of drug, S1) and 2nd (presence of drug, S2) stimulations carried out in fractions 4 and 18. In the control experiments (c), the S2/S1 value was 0.80 ± 0.06. Selegiline exerted a dual effect: it increased electrical stimulation-induced [^3^H]dopamine release in 10^−10^ to 10^−9^ and 10^−6^ to 10^−5^ mol/L concentrations. ANOVA followed by Dunnett’s test, F(9,36) = 3.622, *p* = 0.002, * *p* < 0.05, mean ± S.E.M., n = 4–6. (**B**) The B2/B1 ratio indicates the effect of selegiline on the resting fractional [^3^H]dopamine release determined in fraction 3 (absence of drug, B1) and in fraction 17 (presence of drug, B2). The B2/B1 value was 0.92 ± 0.14 in the control experiments (c). Selegiline added in a concentration range of 10^−13^ to 10^−5^ mol/L was without effect on the resting [^3^H]dopamine release from the rat striatum. ANOVA followed by Dunnett’s test, F(9,36) = 0.581, *p* = 0.803, mean ± S.E.M., n = 4–6.

**Figure 5 ijms-24-13334-f005:**
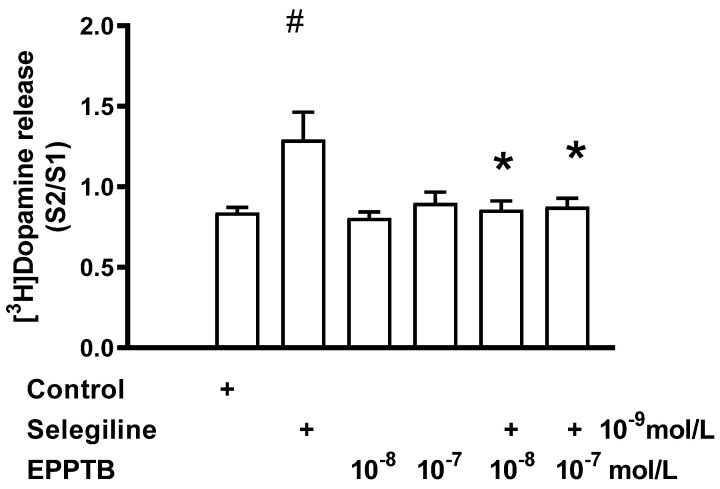
Reversal by EPPTB of the selegiline-induced [^3^H]dopamine release in rat striatum. For experimental procedure, see Figure 2. The resting and the electrical stimulation-induced [^3^H]dopamine release were determined as fractional rates. In these experiments, selegiline was added from fraction 8 to striatal slices in a concentration of 10^−9^ mol/L and maintained throughout the experiment in the presence and absence of EPPTB. When used, EPPTB was added to striatal slices from fraction 1 in a concentration of either 10^−8^ or 10^−7^ mol/L and was present throughout the experiment. One-way ANOVA was followed by Dunnett’s test, F(5,32) = 2.977, *p* < 0.05. Student *t*-statistics for two-means, control vs. selegiline effect, # *p* < 0.01, selegiline vs. selegiline plus EPPTB effects, (EPPTB 10^−8^ mol/L, * *p* < 0.05 and EPPTB 10^−7^ mol/L, * *p* < 0.01), mean ± S.E.M., n = 4–8.

**Figure 6 ijms-24-13334-f006:**
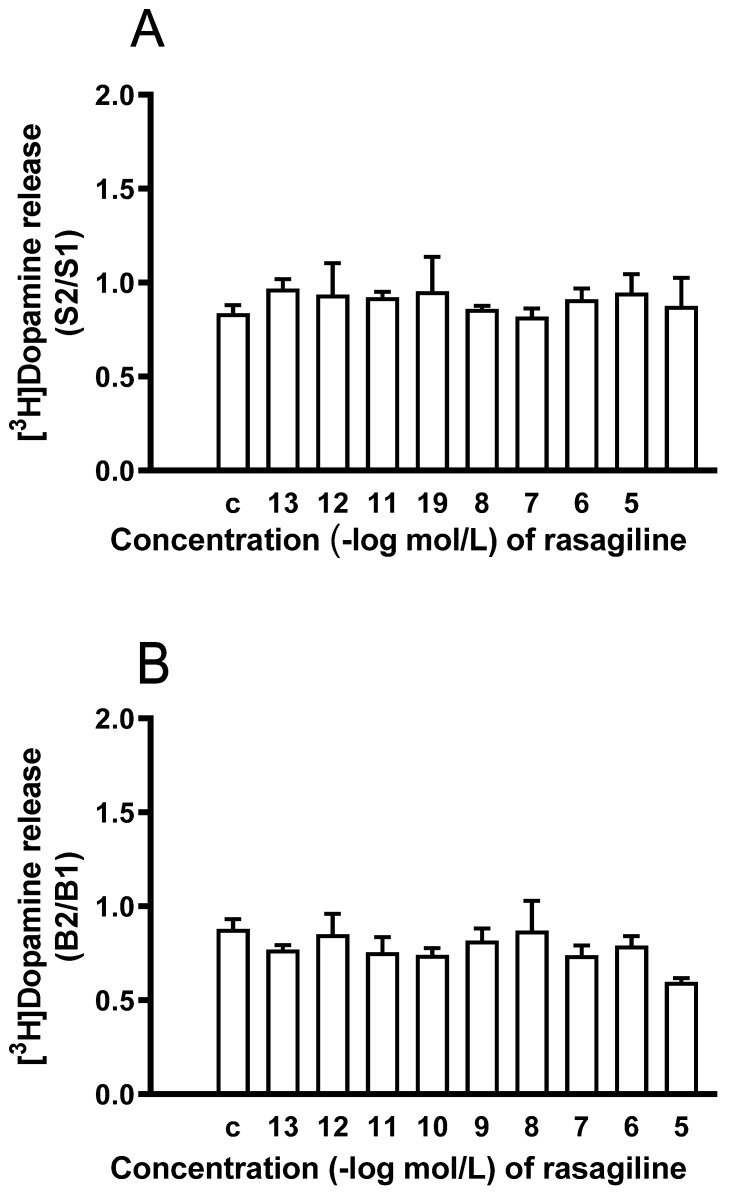
The lack of dopaminergic enhancer effect of rasagiline on the resting and electrical stimulation-induced [^3^H]dopamine release from rat striatum. For experimental procedure, see Figure 2. The resting and electrical stimulation-induced [^3^H]dopamine release was determined as a fractional rate. Rasagiline was added in a concentration range from 10^−13^ to 10^−5^ mol/L to the superfusion buffer from fraction 8 and maintained throughout the experiment. (**A**) The S2/S1 ratio indicates the effect of rasagiline on the electrical stimulation-induced [^3^H]dopamine release determined in the 1st (absence of drug, S1) and 2nd (presence of drug, S2) stimulations carried out in fractions 4 and 18. The S2/S1 ratio in the control experiments (c) was found to be 0.83 ± 0.04. ANOVA followed by Dunnett’s test, F(9,30) = 0.197, *p* = 0.992, mean ± S.E.M., n = 4. (**B**) The B2/B1 ratio indicates the effect of rasagiline on the resting fractional [^3^H]dopamine release determined in fraction 3 (absence of drug, B1) and in fraction 17 (presence of drug, B2). The B2/B1 value was 0.88 ± 0.05 in the control experiments (c). ANOVA followed by Dunnett’s test, F(9,30) = 1.357, *p* = 0.250, mean ± S.E.M., n = 4.

**Figure 7 ijms-24-13334-f007:**
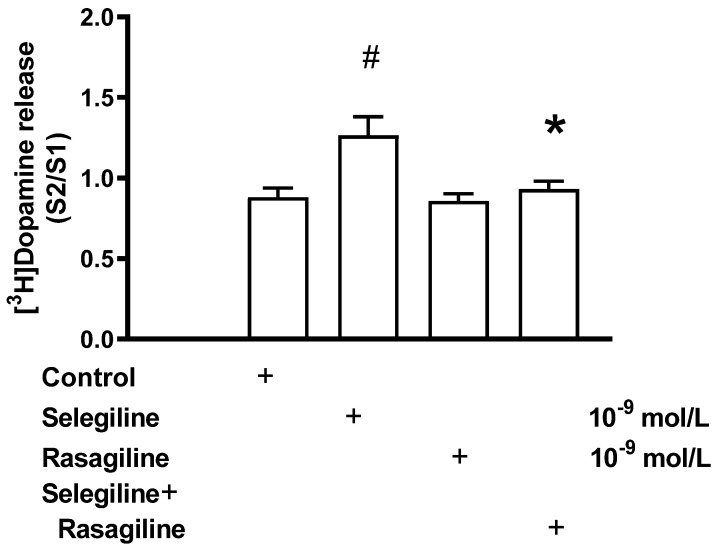
Rasagiline reversed the selegiline-induced [^3^H]dopamine release in rat striatum. For experimental procedure, see Figure 2. The resting and electrical stimulation-induced [^3^H]dopamine release was determined as a fractional rate. Selegiline was added to striatal slices from fraction 8 in a concentration of 10^−9^ mol/L and maintained throughout the experiment in the presence and absence of rasagiline. When used, rasagiline was added to the striatal slices from fraction 1 in a concentration of 10^−9^ mol/L and maintained throughout the experiment. ANOVA followed by Dunnett’s test, F(3,28) = 7.228, *p* < 0.001. Student’s *t*-statistics for two-means, control vs. selegiline effect, # *p* < 0.01, selegiline vs. selegiline plus rasagiline effects, * *p* < 0.05; rasagiline vs. rasagiline plus selegiline effects did not differ significantly, mean ± S.E.M., n = 8.

**Figure 8 ijms-24-13334-f008:**
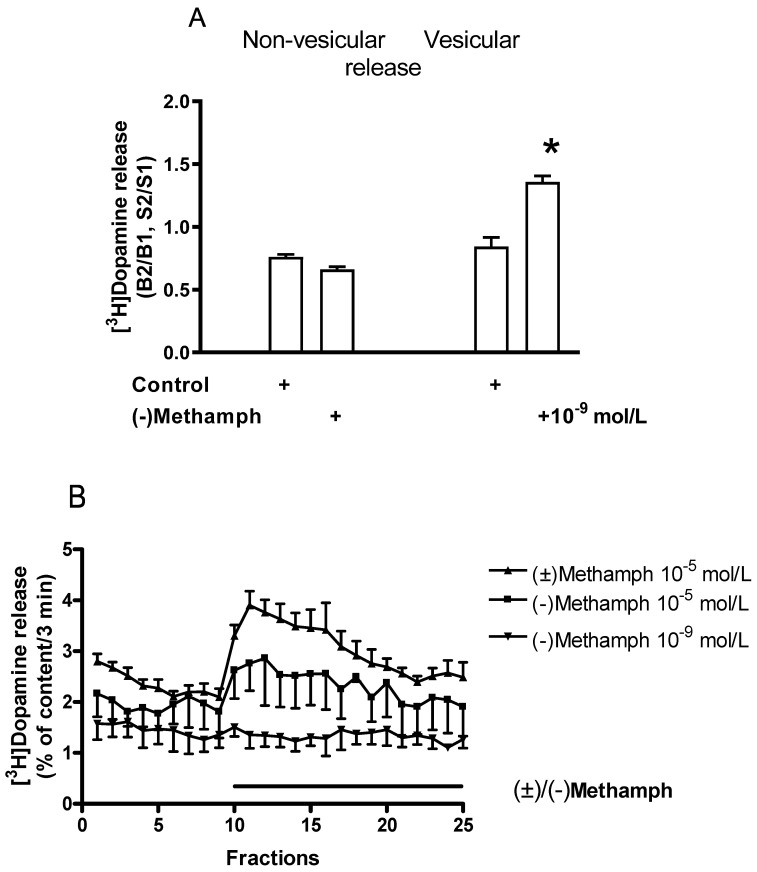
The effects of the despropargyl metabolite of selegiline (−)methamphetamine ((±)/(−)methamph) on [^3^H]dopamine release from rat striatum. For experimental procedure, see Figure 3 and Figure 4. The resting and the electrical stimulation-induced [^3^H]dopamine release were determined as fractional rates. (**A**) (−)Methamphetamine increased vesicular (electrically induced, S2/S1) but not the non-vesicular (resting, B2/B1) [^3^H]dopamine release when added to the striatal slices in a concentration of 10^−9^ mol/L, Student’s *t*-statistics for two-means, * *p* < 0.01, mean ± S.E.M., n = 4–4. (**B**) The effects of (−) and (±)methamphetamine in 10^−5^ mol/L concentration on resting [^3^H]dopamine release in rat striatum. As expected, the [^3^H]dopamine-releasing effect of (−)methamphetamine was substantially less than that of the racemic form. Striatal [^3^H]dopamine release was 4.49 ± 0.60 and 13.05 ± 1.65 percent of the content released in response to 10^−5^ mol/L (−)methamphetamine and (±)methamphetamine, respectively. Student’s *t*-statistics for two means, *p* < 0.01, mean ± S.E.M., n = 4. (−)Methamphetamine added in a concentration of 10^−9^ mol/L was without effect on [^3^H]dopamine release.

**Figure 9 ijms-24-13334-f009:**
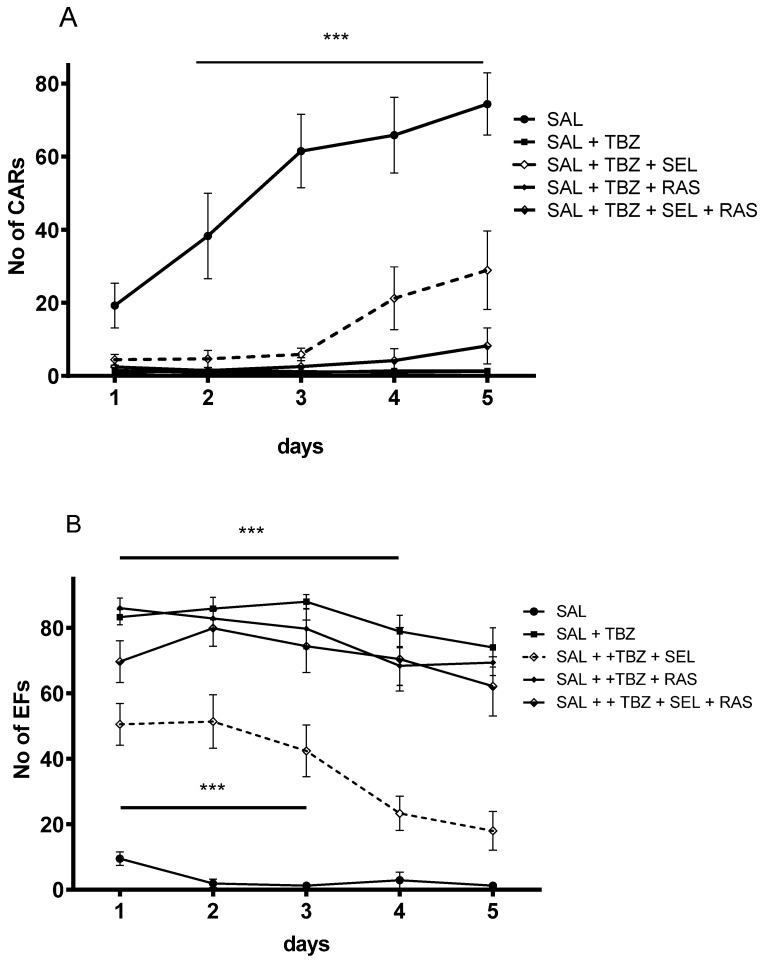
Effect of selegiline, rasagiline, and their combination against tetrabenazine-induced memory loss, measured in a shuttle box. (**A**) panel—number of conditioned avoidance responses (CAR), (**B**) panel—number of escape failures (EF). Tetrabenazine (1 mg/kg) abolished the positive CARs ((**A**) panel) and the escape responses, resulting in about 80% EF ((**B**) panel). This effect was reduced by selegiline (0.001 mg/kg equivalent to 5.33 nmol/kg, sc.) but not rasagiline. The combination of rasagiline with selegiline, however, terminated selegiline’s effect. TBZ, tetrabenazine; SEL, selegiline; RAS, rasagiline. Statistical analysis—CAR - F = 53.86 *** *p* < 0.001, EF F = 73.78 *** *p* < 0.001. No of animals, 8 per group.

**Figure 10 ijms-24-13334-f010:**
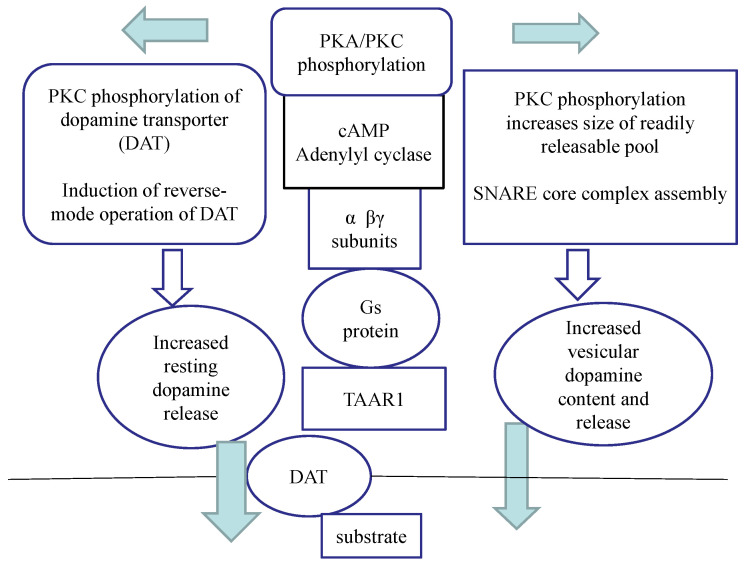
A model for the role of the trace amine associated receptor 1 (TAAR1) and signaling in the mechanisms of selegiline and rasagiline to influence dopaminergic neurotransmission in rat striatum. We propose that the dopaminergic enhancer activity effect of selegiline is related to the stimulation of TAAR1 and its signal transduction pathway. Selegiline is taken up by the plasma membrane dopamine transporter (DAT) and activates TAAR1, a Gs protein-coupled, intracellularly located metabotropic receptor [33,34]. Stimulation of this receptor will result in adenylyl cyclase activation, followed by an increase in cAMP production and augmented intracellular phosphorylation. Protein kinase C (PKC) phosphorylates a series of proteins involved in the exocytotic processes [35,36,37], and the stimulation-evoked vesicular release of dopamine increases. Classical amines, trace amines, and the releaser amphetamines also act as agonists on TAAR1 [19]. Our experiments strongly suggest that the MAO-B inhibitor selegiline might act as an agonist on TAAR1, whereas the MAO-B inhibitor rasagiline suspends selegiline–TAAR1 interaction. Enhanced TAAR1 and signaling may have a beneficial role in the abnormalities in dopaminergic neurochemical transmission observed both in presynaptic and postsynaptic levels [38].

**Figure 11 ijms-24-13334-f011:**
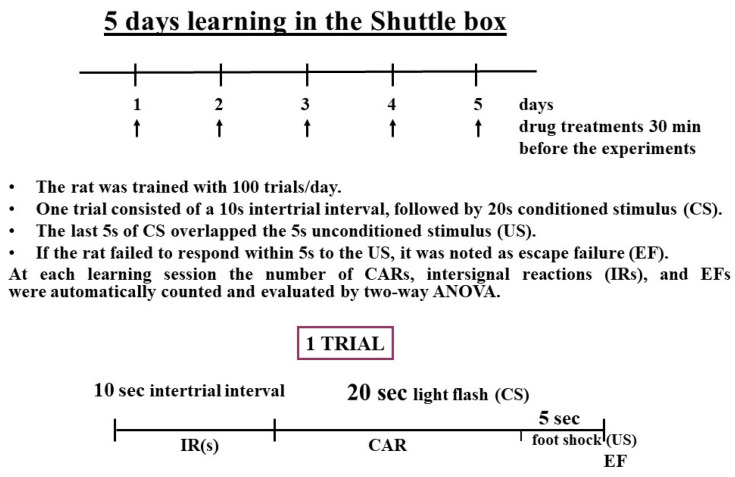
Schematic chart displaying the behavioral experiment design.

## Data Availability

Archived laboratory documents.

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
