# Peer review of "Striking Neurochemical and Behavioral Differences in the Mode of Action of Selegiline and Rasagiline"

_ijms, 2023, doi:10.3390/ijms241713334_

Round 1

Reviewer 1 Report

We consider the manuscript very interesting and pertinent to the readers of this Journal’s Section of Molecular Neurobiology.

The study is the continuation of the previous work of the team, relating the MAO inhibitors pharmacology, where the authors found new shreds of evidence that selegiline possesses a dopaminergic activity enhancer effect, in addition to inhibition of MAO B, and the two effects are independent.

The objectives of this work are clearly stated and comprehensively justified, but the limitations of the study are lacking.

The introduction perfectly integrates the theme's main aspects.

The references used throughout the entire manuscript are old (1 ref. of 2022 and zero ref. of 2023, 2021, and 2020) please, dear authors, include the updated aspects on the subject.

This article is well written, with a good organization of the contents.

Regarding the discussion of the results, we found it suitable. The nice graphic pics/graphs with proper statistical support accompanying the discussion increase the understanding of the discussed theme and clarifies the reasoning. We congratulate the authors on that.

Author Response

REVIEWER 1

  • We indicated the limitation of our study in the text (page 15, line 12)
  • We include additional novel references into the manuscript (Reference Number: 4, 9, 25, 43)

Reviewer 2 Report

In the manuscript titled Striking Neurochemical and Behavioral Differences in the Mode of Actions of Selegiline and Rasagiline, the authors performed ex vivo detection of [3H] dopamine release in rat striatum and demonstrated the neurochemical effect of selegiline and rasagiline. This study contains some interesting findings and is valuable for understanding the role of selegiline and rasagiline in the treatment of Parkinson’s disease. However, some information is missing which makes it hard for people to get the main idea of this manuscript. Therefore, some revisions need to be done before this manuscript could be accepted for publication in the International Journal of Molecular Sciences.

There are some problems in the structure of the abstract, making it hard to grasp the main idea of this article. When I first read it, I thought these words is part of the background: line 9-14, “Selegiline also possessed dopaminergic enhancer effect…”. However, after reading the result part, I think those words are the description of the results of this manuscript. Besides this, the aim and method are not clearly mentioned in the abstract, or maybe they are described in a background way. Please consider rewriting the abstract, in order to helping readers get the idea more easily.

The authors used two different unit in explaining the role of selegiline and rasagiline. For dopamine release from rat striatal slices, picomolar concentrations (10-10-10-9 mol/L) are used; for enhancer dose, gram in weight unit (0.001 mg/kg) is used. Please add some information (such as what mole concentration for enhancer dose) for readers to interpret these changes and results.

For the behavior experiment, which should be a highlight in this manuscript. Please give a schematic chart to display the behavior experiment design. Also in the introduction part, please give some background information of the behavior experimental design and the significance of adding this part.

The figures are not well organized in this manuscript. They displayed the results in an unprofessional way. Some sub-figures are not compact, such as Figure 1 and Figure 3. In Figure 3B, it’s hard for the reader to catch the trend of each subject, as the symbols are overlapped and it’s hard to read each symbol. In Figures 4 & 6, the bar plot can be improved such as cutting off the empty area of the x-axis or enlarging the area of each bar. In Figures 5 & 7, the letter size of each treatment can be reduced and more compact to the axis.

Round 2

Reviewer 2 Report

All other revisions are good to me, but I am still unsatisfied with the figure quality in Figures 4-7 and 8A. As I mentioned in the last comment, "In Figures 4 & 6, the bar plot can be improved such as cutting off the empty area of the x-axis or enlarging the area of each bar. In Figures 5 & 7, the letter size of each treatment can be reduced and more compact to the axis." But I didn't see any improvement on this and no explanation from the authors' side. This may be something wrong when converting PDF to match the publishable format, so I leave this to the editor.

Author Response

Dear Reviewer,

We corrected the text of the manuscript and figures according to your suggestions and uploaded the revised version to the Editorial Office.

We appreciate your effort and work to improve our manuscript.

Kind regards,

Ildiko Miklya